# Theophylline Extracted from Fu Brick Tea Affects the Metabolism of Preadipocytes and Body Fat in Mice as a Pancreatic Lipase Inhibitor

**DOI:** 10.3390/ijms23052525

**Published:** 2022-02-25

**Authors:** Tian-Tian Liu, Xiao-Tian Liu, Gui-Li Huang, Long Liu, Qing-Xi Chen, Qin Wang

**Affiliations:** 1Science Center for Future Foods, Jiangnan University, Wuxi 214122, China; liutiantian@jiangnan.edu.cn (T.-T.L.); longliu@jiangnan.edu.cn (L.L.); 2Key Laboratory of Carbohydrate Chemistry and Biotechnology, Ministry of Education, Jiangnan University, Wuxi 214122, China; 3School of Life Sciences, Xiamen University, Xiamen 361005, China; 21620181153704@stu.xmu.edu.cn (X.-T.L.); chenqx@xmu.edu.cn (Q.-X.C.); 4Agricultural Product Storage and Processing Laboratory, Suzhou Academy of Agricultural Sciences, Suzhou 215105, China; huanggl2015@163.com

**Keywords:** obesity, lipase inhibitors, theophylline, Fu Brick tea, preadipocytes

## Abstract

The dramatic increase in obesity is putting people under increasing pressure. Lipase inhibitors, as a kind of effective anti-obesity drug, have attracted more and more researchers’ attention in recent years because of their advantages of acting on the intestinal tract and having no side effects on the central nervous system. In this study, lipase inhibitor Fu Brick Theophylline (FBT) was screened based on enzyme molecular dynamics, and the inhibition mechanism of lipase inhibitors on obesity was analyzed and discussed at the cellular level and animal model level. We found that FBT had high inhibition effects of lipase with an *IC*_50_ of 1.02~0.03 μg/mL. Firstly, the laboratory used 3T3-L1 proadipocytes as models, flow cytometry was used to detect the effects of FBT on the cycle, apoptosis and intracellular ROS activity of proadipocytes. To study the contents of triglyceride, total cholesterol, related metabolites and related gene and protein expression in adipocytes. The results showed that FBT could reduce ROS production and inflammatory factor mRNA expression during cell differentiation. Secondly, by establishing the animal model of high-fat feed ob nutritional obese mice, the morphological observation and gene expression analysis of body weight, fat rate, adipocyte and hepatocyte metabolism of FBT obese mice were further discussed. It was proven that FBT can effectively reduce the degree of fatty liver, prevent liver fibrosis and fat accumulation, and improve the damage of mitochondrial membrane structure. This study provides a theoretical basis for the screening and clinical treatment of lipase inhibitors.

## 1. Introduction

Adipose tissue is the central organ that maintains homeostasis, and white adipose tissue (WAT) is an important part of the body that stores fat and regulates energy metabolism, but the excessive accumulation of WAT leads to the development of obesity [1]. In WAT, lipid synthesis is directly positively correlated with decomposition efficiency [2]. Many studies have shown that the secretion and expression of various inflammatory factors increase in the white fat of type 2 diabetes patients. These inflammatory factors and inflammatory markers can cause insulin resistance by interfering with normal insulin signal transduction pathways. Hepatic insulin resistance is an important pathophysiological mechanism of glucose and lipid metabolism disorder, which is of great significance for the development of non-alcoholic fatty liver disease (NAFLD) [3].

Lipase plays a key role in human fat metabolism. It breaks down the oil in food into small molecules of glycerol and fatty acids that the body can absorb and metabolize [4]. Screening of pancreatic lipase inhibitory active ingredients has become a research hotspot [5]. In previous studies of pancreatic lipase inhibitors, most inhibitors have shown an effect of improving fatty liver damage and reducing blood lipid levels in the body [6]. Clinical trials have also shown that pancreatic lipase inhibitors can accelerate the emptying of food in the stomach and improve fat metabolism in the body [7]. Chinese herbal medicines are of interest due to their diverse structure, nontoxicity and wide range of sources, and their flavonoids, alkaloids, polyphenols, terpenes and other components show lipase inhibitory effects. At present, studies on the regulation of the metabolism of Fu Brick tea focus only on the effects of polysaccharide extraction on intestinal microorganisms.

As one of the three health drinks, tea enjoys a good reputation in the world. Fu Brick tea is a unique black tea and the most typical tea naturally enriched with selenium which has effects, such as lowering blood sugar and blood fat, anti-oxidation, bacteriostasis and increasing resistance to healthcare effects. In the study of Liu et al., the extraction of polysaccharides from Fu Brick tea improved metabolic disorders by regulating the intestinal microbial population [8]. Fu Brick tea is mainly digested by intestinal flora and has a significant healthcare effect [9]. However, there has been no report on theophylline extracted from Fu Brick tea. In this study, the inhibitory effects and mechanisms of FBT on lipase and the proliferation and differentiation of preadipocytes were studied. The results showed that FBT inhibited the generation of lipid droplets by affecting the differentiation of preadipocytes. Furthermore, FBT can reduce the release of preadipocyte ROS and inflammatory cytokines. At the same time, mice treated with FBT had significantly improved fatty liver and lipid metabolism due to pancreatic lipase inhibition and reduced lipid intake.

## 2. Result

### 2.1. The Molecular Structure of FBT

The 98.78% pure of FBT was obtained with a series of specific separation and extraction steps, which are shown in the Appendix A. The structural formula of FBT is shown in Figure 1. In order to characterize the group information of FBT, nuclear magnetic resonance analysis (1H-NMR and 13C-NMR) was carried out on FBT. The absorption peak frequency on the NMR spectrum, namely chemical shift δ, is one of the important parameters of the NMR spectrum. Figure 1A shows the 1H-NMR analysis of Fu Brick tea pigment, Δ δ3.17 (3H, s, N1-CH3), Δ δ3.34 (3H, s, N3-CH3), Δ δ3.81 (3H, s, N7-CH3), Δ δ7.92 (1H, s, C8-H) and the 1H-NMR peaks Δ 2.67 (3H, s, N1-CH3), Δ 2.67 (3H, s, N3-CH3), δ3.93 (3H, S, N7-CH3), δ8.05 (1H, S, C8-H) belong to similar positions. In the analysis results, a relatively wide peak at Δ 4.2 position is water peak, which may be due to the deviation of peak shifts of other groups. Figure 1B shows 13C-NMR analysis and judgment. The positions of the absorption peaks of Δ 27.94 (n1-CH3), Δ 29.81 (n3-CH3), Δ 33.57 (n7-CH3), Δ 107.35 (C5), Δ 143.27 (C8), Δ 148.31 (C4), Δ 151.55 (C2), Δ 154.79 (C6) are basically consistent with the positions of the carbon spectrum peaks of theanine in the document. From this, it is judged that this compound is tea alkali in Fu Brick tea, and its chemical structure is shown in Figure 1B. In this paper, it is named Fu Brick Theophylline, abbreviated FBT.

### 2.2. Inhibitory Effect and Mechanism of FBT on Lipase

#### 2.2.1. Effect of FBT on Lipase Activity

As shown in Figure 2A, the relative activity of lipase showed a significant downward trend with increasing FBT concentration, and its *IC*_50_ was 1.02 ± 0.03 μg/mL. The relationship between the remaining enzyme activity and enzyme concentration in the presence of different concentrations of FBT was a family of straight lines that all passed through the origin, indicating that it is the reversible property of the inhibition. According to the Lineweaver-Burk method of double reciprocal mapping, a set of straight lines intersecting in the second quadrant was obtained (Figure 2B). FBT increases the *K*_m_ value of the enzyme and decreases the *V*_m_ value, indicating that the type of inhibition on lipase is mixed. The calculated data are summarized in Table 1.

#### 2.2.2. Fluorescence Quenching of FBT on Lipase

As shown in Figure 2C-a, with the continuous addition of FBT, the fluorescence emission peak of lipase gradually decreased but did not affect the peak shape and peak position of the enzyme fluorescence emission spectrum. This shows that FBT and lipase form an enzyme-FBT complex, which prevents the enzyme from binding to the substrate and achieves the effect of inhibiting the catalytic activity of the enzyme. Figure 2C-b is the peak height value corresponding to Figure 2C-a. The peak of the lipase fluorescence spectrum gradually decreases with the increase in the amount of FBT.

According to the Stern–Volmer equation: F_0_/F = 1 + *K*_SV_ [I], with F_0_/F as Y and [I] as X (Figure 2C-c), we can calculate *K*_SV_ = 2.39 × 102 M^−1^ > 100 M^−1^. Therefore, the quenching process of lipase protein molecules by FBT is static. In this case, the binding constant *K*_A_ and the binding site n can be calculated according to the Scatchard equation lg [(F_0_ − F)/F] = lg*K*_A_ + nlg [I]. In Figure 2C-d, using lg [(F_0_ − F)/F] as the *Y*-axis and lg [I] as the *X*-axis, the values of n and *K*_A_ were obtained according to the slope and intercept of the obtained linear equation. The calculated data are listed in Table 2.

#### 2.2.3. Molecular Simulation Docking

The molecular mechanism of lipase inhibition by FBT was studied by the molecular simulation docking software MOE, as shown in Figure 2D. FBT directly interacts with the lipase in the carbonyl oxygen of the carbon 2 bond and has a strong docking force with the Tyr195 residue of the lipase, and the Ser237 Glu220, Asp238, Pro194, Ile193, Arg196, and Thr236 residues also interact with the lipase. FBT has a relatively large molecular size, a part of which is bound to the cavity and a part of which is bound to the outside of the cavity, and the mixed type inhibits the steric hindrance of the substrate and the enzyme active center, which is consistent with the enzyme kinetic analysis [10].

### 2.3. Effects of FBT on 3T3-L1 Preadipocytes

#### 2.3.1. Effects of FBT on 3T3-L1 Preadipocyte Toxicology and Proliferation

The lipid-lowering effects of lipase inhibitors were verified at the cellular level. Changes in the number and volume of fat were determined by the proliferation, differentiation and apoptosis of proadipocytes. Therefore, it is necessary to study the effects of FBT on the proliferation, differentiation and apoptosis-related regulatory genes and proteins of proadipocytes. The A4 quadrant represents the number of apoptotic cells, and the A2 quadrant represents necrotic cells in the flow cytometer. In Figure 3A-(a–e), the results show that FBT caused only a small amount of 3T3-L1 preadipocytes to enter early apoptosis when the concentration was greater than 50 μg/mL, indicating that it has inhibitory effects on preadipocytes but does not cause cell apoptosis and necrosis. In addition, FBT had no significant difference in lactic dehydrogenase (LDH) release from 3T3-L1 preadipocytes (F (1, 4) = 2.000, *p* = 0.2302), indicating that FBT is not toxic to 3T3-L1 preadipocytes (Figure 3A-f). The results show that FBT can inhibit the proliferation of 3T3-L1 preadipocytes, as shown by the morphological observation in Figure 3B-(a–f) and MTT assay in Figure 2B-g. According to the two-way ANOVA, the treatment time (F (5, 48) = 60.22, *p* < 0.0001) and the concentration of FBT (F (1, 48) = 6.119, *p* = 0.0170) had a significant effect on the proliferation of 3T3-L1 preadipocytes, and the two factors had a significant correlation (F (5, 48) = 2.757, *p* = 0.0287). The average inhibition rates were 35.2% and 51.3% after treatment for 24 h and 48 h, respectively. The decrease in the number of 3T3-L1 preadipocytes is due to FBT inhibiting proliferation.

#### 2.3.2. Effects of FBT on the Cell Cycle of 3T3-L1 Preadipocytes

We used different concentrations of FBT (0, 5, 10, 25, 50, and 100 μg/mL) to simultaneously treat logarithmic phase preadipocytes for 48 h, and the results are shown in Figure 3C. With increasing FBT concentration, the proportion of 3T3-L1 preadipocytes entering the S phase decreased, and the proportion of cells in the G2/M phase increased. The G1 phase stabilized after decreasing. Further cell cycle analysis revealed that the cells were blocked in the G2 phase, which was the reason why FBT affected preadipocyte proliferation.

#### 2.3.3. Effects of FBT on ROS Levels in 3T3-L1 Preadipocytes

The DCFH-DA was used as a fluorescent probe of the active oxygen detection kit to combine the active oxygen in the cell with the fluorescent DCFH to generate fluorescent DCF. The level of ROS was measured on a flow cytometer based on the amount of DCF fluorescence. Figure 3D-a shows the ROS peak of adipose cells 24 h after FBT treatment. With the increase of FBT concentration, the mean value of ROS X gradually decreases compared with the control group (3D-c), and the peak value gradually shifts to the left (3D-b). These results suggest that FBT can reduce the ROS level of 3T3-L1 preadipocytes (F (5, 12) = 15.48, *p* < 0.0001). Furthermore, the mRNA expression levels of ROS-related genes, including the key kinase interleukin 6 (IL-6) (F (3, 4) = 126.5, *p* = 0.0002), tumor necrosis factor (TNF-α) (F (3, 4) = 457.5, *p* < 0.0001), and inhibitor kappa B kinase β (IκKβ) (F (3, 4) = 360.6, *p* < 0.0001), were downregulated with increasing concentrations of FBT, as shown in Figure 3D-d. TNF-α and IL-6 have a variety of effects on the physiological functions of adipocytes, and IKK-β is a key hub for inflammatory signals to interfere with insulin signals [11], which can mediate chronic inflammation induced by obesity. Their decrease indicates that FBT has a certain restriction on the inflammatory response.

#### 2.3.4. Effects of FBT on 3T3-L1 Preadipocyte Lipid Production

Lipid droplets store various substances required for energy metabolism and are embedded with various proteins on their surfaces. Triglycerides (TGs) are stored in the cytoplasm of fat cells in the form of lipid droplets, while total cholesterol (TC) exists in the plasma membrane of cells and is the main component of lipid droplets in fat cells. In Figure 4A-a,b, by testing TG and TC in the 100 μg/mL FBT and control groups on different days, it was found that the accumulation of TG and TC in cells treated with FBT began to decrease significantly from the second day of cell differentiation. On the eighth day, TG accumulation decreased by 37.5–37.7% and TC accumulation decreased by 53.7–54.7% compared with the control group. After that, we tested the contents of TG and TC at different concentrations of FBT on the sixth day (Figure 4A-(c,d)). The results showed that FBT reduced the contents of TG (F(5, 6) = 153.6, *p* < 0.0001) and TC (F(5, 6) = 107.2, *p* < 0.0001) in a gradient manner.

The substrate method was used to verify the lipase in cells directly, demonstrating that FBT could inhibit the decomposition of lipase, and FBT also inhibited the hydrolysis of triglycerides in tissues. The results showed that lipase activity decreased in a concentration-dependent manner with drugs (Figure 4B-a), (F(5, 18) = 703.6, *p* < 0.0001). An ATGL kit was used to detect the decreased triglyceride lipase content in fat cells (Figure 4B-b), (F(3, 12) = 83.50, *p* < 0.0001), and lipase activity was collected and detected after cell lysis. The substrate method was used to verify the lipase in cells directly, which proved that FBT could inhibit the decomposition of lipase, and FBT also inhibited the hydrolysis of triglyceride in tissues.

#### 2.3.5. Effects of FBT on the Expression of mRNAs Related to Differentiation and Lipid Production in Preadipocyte 3T3-L1 Cells

After treatment with FBT, some mRNAs encoding fat absorption-related and synthesis-related enzyme genes of preadipocytes were downregulated, as shown in Figure 4C-a, the TG-degradation rate-limiting enzyme lipoglycerol lipase (ATGL) (F (3, 4) = 1206, *p* < 0.0001) and hormone-sensitive lipase (HSL) (F (3, 4) = 196.9, *p* < 0.0001), lipoprotein esterase (LPL) (F (3, 4) = 328.8, *p* < 0.0001) were related to fatty acid absorption, and acetyl-CoA carboxylase (ACC) (F (3, 4) = 221.8, *p* < 0.0001) and fatty acid synthase (FAS) (F (3, 4) = 1375, *p* < 0.0001) were related to fat synthesis. These results indicated that FBT had a certain inhibitory effect on the synthesis of fat. In Figure 4C-b, the mRNA expression levels of fatty acid oxidation rate-limiting enzymes, including carnitine palmitoyl transferase (CPT-1) and acetyl-CoA oxidase (A-COX) genes, were significantly upregulated. These results indicate that FBT may play a certain role in promoting fatty acid oxidation.

Adenosine-activated protein kinase (AMPK) is a crucial energy regulator of cellular anabolism and catabolism. When AMP/ATP levels rise due to cell stress, AMPK will be phosphorylated and then activate multiple downstream target molecules to reduce ATP consumption and inhibit lipid and cholesterol synthesis, etc., [12]. Conversely, the decrease of AMP/ATP will increase the synthesis of ATP and promote the oxidation of fatty acids [13]. Figure 4D-a showed that FBT increased AMPK mRNA expression in adipose tissue. Meanwhile, AMPK and phosphorylated AMPK protein expression were detected (Figure 4D-b), and the difference in AMPK protein expression was not significant, while p-AMPK expression was significantly increased. Our results suggest that FBT is involved in adipose tissue regulation, most likely by activating the AMPK pathway to reduce lipid decomposition and lipid production. At the same time, the expression of P-AMPK was up-regulated, suggesting that the drug might protect mitochondria from ROS oxidative damage by activating AMPK

### 2.4. FBT Reducing Fat Based on an OB Mouse Obesity Model

#### 2.4.1. Effect of FBT on the Accumulation of Lipid Droplets in Non-Alcoholic Fatty Liver

To investigate the therapeutic effects of FBT on diet-induced obesity, we orally administered vehicle (saline) FBT or orlistat to OB/OB mice for 40 days. Although food intake was similar to that in the control group, FBT supplementation significantly alleviated the perirenal fat and hepatomegaly detected by PET-CT in mice (Figure 5A). Figure 5A-a showed that the fat accumulation and fatty liver volume in the control group increased significantly. Compared to the perirenal fat of mice treated with fuzhuancha extract (Figure 5A-b) to that at the beginning of gavage, we found that the perirenal fat decreased, and there was little difference in the size of the liver at the end of gavage and at the beginning of gavage. At the same time, compared with the control group, the volume of perirenal fat and liver fat decreased significantly in the FBT treatment group after gavage, indicating that the fat accumulation in the abdominal cavity and the hypertrophy of fatty liver can be significantly reduced after Fu Brick tea extract treatment. The results in Figure 5B show that the weight of mice in the control group exhibited an upward trend. The average weight of mice in the control group was 57.2 g after 48 days of modeling and 40 days of continued growth. A significant inhibitory effect was observed in all groups administered by gavage. The average weight of mice in the orlistat group was controlled at 64.3 g, and the weight of mice in the FBT group was reduced to 51.8 g, indicating that FBT had a certain inhibitory effect on the weight of obese mice and even had a better effect on fat reduction than the positive control. Strikingly, liver sections stained by Oil Red O staining (Figure 5C) and hematoxylin & eosin (Figure 5D) showed fewer and smaller lipid vacuoles in the liver of the FBT-treated mice than in the untreated mice. The reduced lipid accumulation in the livers of the treated mice was supported by Oil Red O staining and biochemical analysis, which showed an approximate 30% reduction in liver triglyceride (TC) content and a 50% reduction in liver cholesterol (TG) (*p* < 0.05; Figure 5E).

#### 2.4.2. Effect of FBT Remission on Liver Fibrosis and Degree of Injury in Mice

The livers of OB/OB mice in the control group exhibited hepatocytes with lipid droplets spread throughout the cytoplasm and cells with characteristics of myofibroblasts stained blue near the hepatocytes and lipid drops, thereby indicating the development of fibrosis in Masson’s trichrome staining (*p* < 0.01) (Figure 6A). Next, to determine the effects of FBT on glucose homeostasis and insulin sensitivity, GTT and ITT were performed after 40 days of FBT treatment. As shown in Figure 6B, FBT-treated OB/OB mice exhibited lower glucose levels at all time points up to 90 min after intraperitoneal glucose (Figure 6B-a) or insulin injection (Figure 6B-b) compared with control OB/OB mice. The fluctuation of blood glucose levels is small, and the mice in the Fu Brick tea extract group have a higher tolerance to glucose.

However, we found that blood glucose did not decrease but increased rapidly after insulin injection in the control group and orlistat group, which is consistent with the characteristics of type 2 diabetes caused by obesity. The ultrastructural analysis of the liver cells of the OB/OB mice revealed many of the different organelles, such as mitochondria, rough-surfaced endoplasmic reticulum (rER), lipid droplets, lysosomes and nuclei, between the control and treatment groups and the presence of lipid droplets spread throughout the cytoplasm in the control group (Figure 6C-i). In addition, we found that the mitochondrial membrane structure of the control and orlistat groups was damaged, the coloring was deep, and the ridge structure was unevenly distributed (Figure 6C-ii). We found that the apoptotic cells in FBT-treated mouse liver tissues were reduced, and they could be specifically labeled with fluorescence area quantification by TUNEL staining, as shown in Figure 6D. After FBT treatment, the fluctuation in blood glucose levels (4.5 mM/L–6.5 mM/L) in the mice remained relatively small compared with that of the control group both in GTT and ITT. Furthermore, FBT also increased the total antioxidant capacity in tissues and serum detected by the Solarbio kit, as shown in Figure 6E.

#### 2.4.3. Effects of FBT on Lipid Metabolism in Non-Alcoholic Fatty Liver Tissue

Finally, a lipid metabolomics analysis was performed to further verify the lipid inhibition effect of FBT. The metabolomic data is shown in Figure 7. Compared with the FBT-treated group, the contents of 30 triglyceride TAGs in the liver of the control group were significantly higher, and the contents of 10 TAGs (all unsaturated fatty acids) were significantly lower (Figure 7A). At the same time, as shown in Figure 7A,B, the accumulation of a large amount of diglyceride (DAG), fatty acid (FA) and acetylated fatty acid (FAHFA) in the liver tissue of mice without FBT treatment may be due to the inhibition of lipase in the intestine by FBT such that a large number of TAGs cannot be broken down into DAG, FA, and FAHFA and directly excreted. Furthermore, the contents of phosphatidylcholine (PC) and phosphatidylethanolamine (PE) were also clearly different. FBT-treated mice had 20 types of PC and 17 types of PE accumulation, while mice without FBT treatment had only two types of PC and eight types of PE accumulation. PC and PE are often used in the treatment of non-alcoholic fatty liver, which can promote the formation of HDL and promote lipid transport [13].

## 3. Discussion

In this study, a relatively safe and effective lipase inhibitor, FBT was screened from Fu Brick tea, and the total yield was 1.34%. The compound with a molecular weight of 195 was collected by an automatic purification system, and the possible molecular formula was determined by Q-Exactive high-resolution liquid chromatography-mass spectrometry analysis. 1H-NMR and 13C-NMR scanning spectrograms determine that the compound is theophylline. Its *IC*_50_ value on lipase is 1.02 μg/mL, and the inhibitory type is reversible mixed inhibition. The endogenous UV fluorescence results show that it has a static quenching effect on the emission spectrum of lipase, and the binding site n is 1.754, which is close to two sites. The molecular simulation results show that FBT may interact with several amino acids of lipase and directly act on the Tyr195 residue. According to the preliminary judgment of lipase kinetics, this active substance has potential in the study of anti-obesity drugs.

Our results demonstrated that the direct effects of FBT on the differentiation of 3T3-L1 cells were significant. Differentiation of preadipocytes into mature adipocytes is considered an important goal for the development of anti-obesity drugs [14]. The results of flow cytometry were analyzed for the changes of intracellular ROS content after FBT treatment. The reduction of ROS can protect the mitochondria of liver cells from damage, which is consistent with the results of in vivo experiments (Figure 6C). Obesity has also been related to antioxidant defense enzymes. Some findings have suggested that the activity of SOD increases at the onset of obesity development in an attempt to combat the increased generation of free radicals [15]. Therefore, understanding the molecular mechanisms that tightly control adipocyte development and adipogenesis will provide valuable information for controlling obesity [16]. We found that FBT can significantly inhibit the proliferation of preadipocytes, but LDH treatment experiments confirmed that FBT is not toxic to cells. The flow cytometry detection of PI single staining and analysis showed that FBT blocked 3T3-L1 preadipocytes in the G2 phase and affected proliferation. However, as FBT had no apoptotic effect on 3T3-L1 preadipocytes in V-FITC/PI double staining, FBT inhibited the proliferation of cells by affecting cell differentiation rather than inducing cell apoptosis and necrosis. Macrophages that infiltrate fat tissues release cytokines, such as TNF-a, IL-6 and IκKβ. These factors are responsible for obesity-related disorders, including hypertension, diabetes, atherosclerosis, insulin resistance, and non-alcoholic fatty liver disease [17]. The upregulation of cytokines, such as IL-6 and TNF-a in adipose tissue serves as a marker of obese adipocytes [18]. Rodents show an increase in oxidative stress and proinflammatory cytokines (TNF-a, IL-6 and IκKβ) in white adipose tissue after a high-fat diet (HFD) [19]. According to our study, anti-inflammatory effects of FBT have been detected at the cellular level and have not been detected in animals. In the flow cytometer test using the DCFH-DA method, the production of ROS during differentiation in FBT-treated cells was reduced in vitro, while the total antioxidant capacity of white fat, liver tissues and serum was improved in FBT-treated mice induced by a high-fat diet in vivo. Acetyl-CoA carboxylase (ACC) produces malonyl-CoA, which inhibits CPT1 and has been shown to regulate the accumulation of body fat [20]. While inhibiting the synthesis of fat, it promotes the oxidation of fatty acids and the absorption of glucose [21]. We observed a decrease in ACC expression in FBT-treated fat progenitor cells, suggesting that it lessened CPT-1 inhibition and promoted its expression. At the same time, inhibiting lipolysis is more conducive to maintaining a healthy energy metabolism balance. Some data show that inhibition of ATGL and HSL expression and activity can effectively inhibit the lipid reaction and improve metabolic disorders [22,23]. The specific knockout of the ATGL gene in mouse white fat cells not only inhibited lipolysis and decreased blood lipids but also inhibited the expression of genes related to fat absorption and synthesis, enhanced the hepatic insulin signaling pathway, and improved glucose tolerance [24]. LPL is closely related to the degradation of TG and the absorption of FA in cells. Adipose tissue-specific LPL deficiency reduced fat storage in OB/OB mice [25]. Other studies have shown that downregulating the expression of LPL and FAS in white fat can effectively inhibit the hypertrophy of fat cells [26]. The results of this study showed that by downregulating the expression of ACC, FAS, LPL, HSL and ATGL, FBT can inhibit the TG synthesis pathway in fat cells, reduce the synthesis of TG and other lipids, and reduce the release of glycerol and FFAs, thus maintaining the stable state of lipid metabolism.

The build-up of lipids in hepatocytes suggests a possible interference with mitochondrial and microsomal function, leading to an interruption in the transport of lipoproteins and a build-up of fatty acids [27]. Fluorescence staining with Tunnel and electron microscope observation of mouse liver showed that the liver status of mice treated with FBT showed decreased fluorescence spots and the mitochondrial status of mice tended to be normal. This confirms that FBT can protect liver cells from apoptosis and inflammation. Non-alcoholic fatty liver disease often leads to apoptosis and inflammation of liver cells and fibrosis of tissues [28,29]. Initially, characterized by hepatic steatosis and defined as liver fat levels in excess of 5% of the liver’s weight, NAFLD can progress to fatty hepatitis, fibrosis, and cirrhosis [30]. The reduction in liver triglycerides in the FBT-treated mice may be explained by metabolomic analysis, including enhanced hepatic triglyceride secretion, decreased hepatic lipogenesis, enhanced intracellular lipolysis, and increased hepatic fatty acid oxidation. Excessive fat storage in the liver can be caused by a range of metabolic disorders, including defective fatty acid oxidation, enhanced fat production, impaired triglyceride secretion, and increased intake of fatty acids from circulation [31]. We found that FBT treatment lowered hepatic fat accumulation and improved glucose homeostasis in HFD-fed mice. Through Oil Red O staining, we found that the size and number of lipid particles in the liver of FBT-treated mice decreased. Meanwhile, according to observations under transmission electron microscopy, the mitochondria of the control group were seriously damaged, resulting in damage to fatty acid oxidation and accumulation of TAG, DAG FA and FAHFA. A large amount of evidence shows that insulin resistance leads to fatty liver, and fatty liver may also lead to obesity and insulin resistance to the liver, which further leads to a vicious cycle [32]. In obesity and insulin resistance, increased intrahepatic fatty acid inflow from lipolysis of diet or adipose tissue is generally believed to be a major driver of the development of NAFLD [33,34]. We found that FBT could maintain the stability of blood glucose and insulin in mice, which is closely related to the reduction in FA and FAHFA in the liver. In addition, FBT can inhibit lipase in the intestine, and its mechanism is to reduce the accumulation of triglycerides in fat and liver tissues by affecting the absorption of triglycerides in the intestine. In a study by Melha Benlebna et al., excessive accumulation of acetylated fatty acid FAHFA in non-fat tissues was found to often lead to liver damage and fibrosis [35]. Giovanni Solinas et al. found that the accumulation of acetylated fatty acids FAHFA in non-adipose tissues would lead to excessive accumulation of lipids and metabolic disorders [36]. However, we found in the metabolomics analysis that regardless of whether the accumulation of PC and PE in liver tissues increased after FBT treatment, the accumulation of PC and PE was related to the transport of HDL and VLDL, which could increase the transport of cholesterol and triglycerides [37]. Further reducing the accumulation of lipids in the liver, the specific mechanism of increasing the accumulation of PC and PE in liver tissue after taking FBT remains to be studied. According to Kocelak et al. and Tonstad et al., lipase inhibitors can reduce the level of LDL in the blood [38].

In general, the effect of FBT on obese mice verified the results of adipocyte level research analysis. Figure 8 summarized the possible mechanism of FBT for lipid reduction and weight loss. Combined with the results of cell experiments, FBT can reduce the decomposition and synthesis of fat and the accumulation of triglyceride and cholesterol through AMPK activation. At the same time promote the oxidative decomposition of fatty acids so as to achieve the purpose of reducing fat and weight loss. The liver tissue staining and lipid metabonomics analysis showed that FBT could repair nonalcoholic fatty liver injury in obese mice. This provides a potential possibility for natural product inhibitors as anti-obesity drugs and provides new ideas for further research.

## 4. Materials and Methods

### 4.1. Materials and Sample Preparation

Lipase (EC3.1.1.3) from *Mucor miehei* (lyophilized powder) was the product of Sigma-Aldrich. The specific activity of the enzyme was over 4000 U/mg (using olive oil). Please refer to the explanation in the literature for the reason for choosing this lipase [39]. DMEM, fetal bovine serum (FBS), and antibiotics were purchased from Genetimes Technology Co., Ltd. (Shanghai, China). Lipase, 4-nitrophenyl palmitate (4-NPP), DMSO, 3-isobutyl-1-methylxanthine (IBMX), dexamethasone (DEX), insulin, and Oil Red O powder were obtained from Sigma–Aldrich (Shanghai, China).

### 4.2. Extraction of Active Ingredients from Fu Brick Tea

In this study, Fu Brick tea, a traditional Chinese fermented black tea, was used as the material (Purchased from China Shannxi Jingyang YiChangMing Fu Brick tea Co., LTD, Shannxi, China). The tea powder was obtained in a pulverizer, weighed to 500 g with 3 L of 90% ethanol three times for extraction, ultrasonically treated for 30 min each time, concentrated by rotary evaporation under reduced pressure at 45 °C, freeze-dried and stored at −20 °C for later use. After removing impurities in the crude extract with petroleum ether and isovolume extraction with ethyl acetate and n-butanol, most of the active substances with inhibitory effects of lipase existed in the ethyl acetate phase. After further passing through an LH20 gel column and eluting and separating with 25%, 50%, 75% and 100% ethanol, respectively. Pure concentrations of Fu Brick tea extract were obtained, and the single product FBT was analyzed through a high-performance liquid phase. The molecular weight and structure of the single substance were determined through Q-Exactive high-resolution LC/MS and nuclear magnetic resonance, which are methods described in the previous literature [40].

### 4.3. Analysis of Inhibition Kinetics of FBT on Lipase and Molecular Simulation Docking

The inhibition rate, type and mechanism were measured using the method we previously described [41]. The type of lipase inhibition was further explored by FBT. Under the conditions of the most suitable live system, the lipase concentration was fixed (0.2 μg/mL), the concentration of substrate 4-NPP was changed, and the initial reaction rate at different FBT concentrations was determined. Endogenous fluorescence spectroscopy was used to analyze the interaction between FBT and lipase. First, a fluorescence scan was performed on 2 mL of 0.1 mg/mL lipase, followed by the addition of an interval of 2 μL of FBT inhibitor for each data point. The docking simulation of the enzyme molecule and the effector was conducted using the bioinformatics analysis software MOE (Molecular Operation Environment). The energy of the enzyme molecule and the effector was minimized, and the docking site with the highest score was found. The docking parameter setting was according to Chen [42].

### 4.4. Cell Culture

3T3-L1 preadipocytes were purchased from Shanghai Fuheng Biotechnology Co., Ltd. (Shanghai, China). Cells were maintained in DMEM supplemented with 10% FBS and 1% antibiotics in an atmosphere of 5% CO_2_ at 37 °C. Adipocyte differentiation was induced by treating cells for 48 h in media containing 10% FBS, 0.5 mM IBMX, 1 µM DEX, and 10 µg/mL insulin. The medium was replaced with medium supplemented with only 5 µg/mL of insulin every other day. The cells were treated with or without FBT for 8 days during adipogenesis.

#### 4.4.1. Effects of FBT on the Proliferation of 3T3-L1 Preadipocytes

3T3-L1 preadipocytes in the logarithmic growth phase were inoculated into 96-well plates at 2 × 10^3^/well and cultured for 48 h after adherence to media containing different concentrations of FBT. Four parallels were used to determine the absorbance A at 570 nm of each treatment group by MTT assay. Value-added rate% = Sample Group A_570_/Control Group A_570_ × 100%. A lactate dehydrogenase kit was used to detect cytotoxicity.

#### 4.4.2. Determination of Intracellular TC/TG

During the differentiation process on Days 2, 4, 6, and 8, the effects of different concentrations of FBT on preadipocyte differentiation and lipid production were detected. After digestion with trypsin, a certain amount of isopropanol was added for ultrasonication for 1 h to help dissolve intracellular lipids and then centrifuged at 3000 rpm for 10 min. The total triglyceride (TC) and total cholesterol (TG) of the supernatant were measured in accordance with the assay kits (Solarbio).

#### 4.4.3. Determination of Glucose Consumption and Adiponectin Production in 3T3-L1 Preadipocytes

Preadipocytes were divided into a control group, model group and dose group. The control group was differentiated into a normal medium; 1 μM DEX and 5 μg/mL insulin were added to the model group; and the dose group was divided into four concentrations of FBT: 10, 20, 100 and 200 μg/mL. The medium was changed every 2 days. After differentiation, the cells were collected, and the supernatant was collected after digestion and lysis. The glucose oxidase assay kit (Pulilai Gene Technology Co., Ltd.) was used to measure the glucose consumption of each concentration group and determine whether the modeling was successful. A mouse adiponectin enzyme-linked immunosorbent assay (ELISA) detection kit (QinCheng Biological) was used to determine the adiponectin content in the cell supernatant.

#### 4.4.4. Cell Cycle Analysis and Determination of ROS Production by Flow Cytometry

The cell cycle was detected with PI single-staining Annexin, apoptosis and necrosis were detected with Ⅴ-FITC/PI staining, and intracellular reactive oxygen species (ROS) production was detected with the DCFH-DA method; all were detected using a flow cytometer and performed as previously described [43]. All of the reagents above were purchased from Beyotime Biotechnology (Shanghai, China).

### 4.5. RNA Analysis

Total RNA from tissues and cells was isolated using TRIzol (Sigma-Aldrich, Shanghai, China), and cDNA was synthesized using HiScript II Q Select RT SuperMix (Vazyme, Nanking, China) for qPCR. Quantitative PCR was performed with ChamQ Universal SYBR qPCR Master Mix (Vazyme, Nanking, China) using a ROCHE Light Cycler 96 (ROCHE, Switzerland). The primer sequences are shown in Appendix A.

### 4.6. Animals

The animal experiment was conducted in accordance with the “Laboratory Animal Guideline for Ethical Review of Animal Welfare” (GB/T 35892-1272018) and the Experimental Animal Management and Ethics Committee of Xiamen University (Certificate no. SYXK2013-0006). Five-week-old male OB/OB mice were obtained from GemPharmatech Co., Ltd. (Nanjing, China) and maintained on a chow diet for 1 week with a 12 h light/dark cycle for acclimatization. After adaptation to the experimental environment for 3 days, the mice were fed a HFD (ResearchDiets, #D12492, New Brunswick, NJ, USA) for 10 weeks. The HFD provided 5.21 kcalg^−1^ of energy (20% calories from protein, 60% calories from fat, and 20% calories from carbohydrate). The mice were randomly divided into three groups (*n* = 6 for each of the treatment groups): the control group (gastric gavage normal saline), FBT treatment group (gastric gavage 60 mg/kg FBT), and orlistat treatment group (gastric gavage 800 mg/kg orlistat). Bodyweight was measured weekly, and food intake was recorded every other day. We used PET-CT to evaluate the effect of FBT on perirenal fat and fatty liver hypertrophy in mice. Glucose and insulin tolerance tests were performed according to a previous paper [44].

#### 4.6.1. Preparation of Liver, Serum and White Fat Tissue Samples

All mice were sacrificed on the 42nd day of intragastric administration. The liver and white fat tissue from the entire brain tissue were immediately separated, and the samples were stored in AllProtectTM (Beyotime Biotechnology, Shanghai, China). The hippocampal tissues were used for other experiments.

#### 4.6.2. Oil Red-O Staining

Mouse adipocytes were fixed with 10% formalin solution for 10 min and stained with freshly prepared Oil Red O solution for 30 min at room temperature. Intracellular lipid contents were measured by extracting Oil Red O with isopropanol, and the absorbance was measured at 500 nm with Thermo Multiskan FC (Shanghai, China). Liver tissue was embedded in OCT after dehydration with 30% sucrose. The frozen sections were operated according to the Oil Red O staining kit purchased from Solarbio. Images were acquired using a Leica DM 4B microscope (Wetzlar, Germany).

#### 4.6.3. Staining of Liver and White Adipose Tissue

Liver and white adipose tissue were dehydrated and sectioned in a routine procedure. Liver and adipose tissue were stained with an HE staining kit. In addition, a Masson staining kit (Beijing Solarbio Science & Technology Co., Ltd., Beijing, China) and a TUNEL fluorescence staining kit (Sangon Biotechnology Co., Ltd. Shanghai, China) was used to evaluate the degree of liver injury. Images were acquired using a Leica DM 4B microscope (Wetzlar, Germany).

#### 4.6.4. Transmission Electron Microscopy

For each group, three animals were analyzed. The liver fragments were fixed overnight in a solution containing 2.5% glutaraldehyde and 4% formaldehyde in 0.1 M cacodylate buffer. The method of transmission electron microscopy was as described in the previous literature [27].

#### 4.6.5. Metabolomics Analysis

The liver lipid LC-QTOFMS-based metabolomic experiments and data analysis were commissioned by Shanghai BIOTREE Biotechnology Co., Ltd.

### 4.7. Statistical Analysis

The data are presented as the mean ± SD. Differences between the means of individual groups were assessed by one-way analysis of variance followed by a multiple comparisons test; differences were considered significant at *p* < 0.05. The statistical software package Prism6.0 (GraphPad Software, LaJolla, CA, USA) was used for these analyses.

## 5. Conclusions

In summary, FBT, a highly effective lipase inhibitor, was extracted from Fu Brick tea. The extraction process is convenient and feasible. The inhibition mechanism of FBT on lipase was explored from the perspective of enzyme molecular dynamics. The *IC*_50_ value of FBT on lipase was 1.02 ± 0.03 μg/mL, which was a reversible mixed inhibition. At the same time, at the cellular and animal experiment levels, FBT downregulated the mRNA expression of ACC, FAS, LPL, HSL, ATGL, upregulated the mRNA expression of A-COX and CPT-1, inhibited the synthesis and breakdown of TG and TC in 3T3-L1 adipocytes, promoted fatty acid peroxidation, and maintained a stable state of lipid metabolism. Surprisingly, FBT also increased sugar consumption and promoted adiponectin activity in the IR model, inhibiting differentiation, eliminating inflammatory factor proinflammatory cytokines (TNF-a, IL-6 and IκKβ) and reducing ROS production. In the animal experiments, we confirmed that FBT can effectively alleviate fatty liver levels, prevent liver fibrosis and accumulation of lipid droplets, reduce liver cell apoptosis, improve damage to mitochondrial membrane structure, and ensure normal β-oxidation. Furthermore, FBT reduces the accumulation of TG and TC in the liver and increases the total antioxidant capacity of the tissues as well as the metabolism of DAG, FA and FAHFA. In this study, FBT was first applied to fat cells and obese mice. The inhibition effect and safety were evaluated, and the lipid-lowering mechanism of FBT influencing fat metabolism was preliminarily discussed. This study provides a new idea for natural product inhibitors as anti-obesity drugs and provides a theoretical basis for clinical treatment.

## Figures and Tables

**Figure 1 ijms-23-02525-f001:**
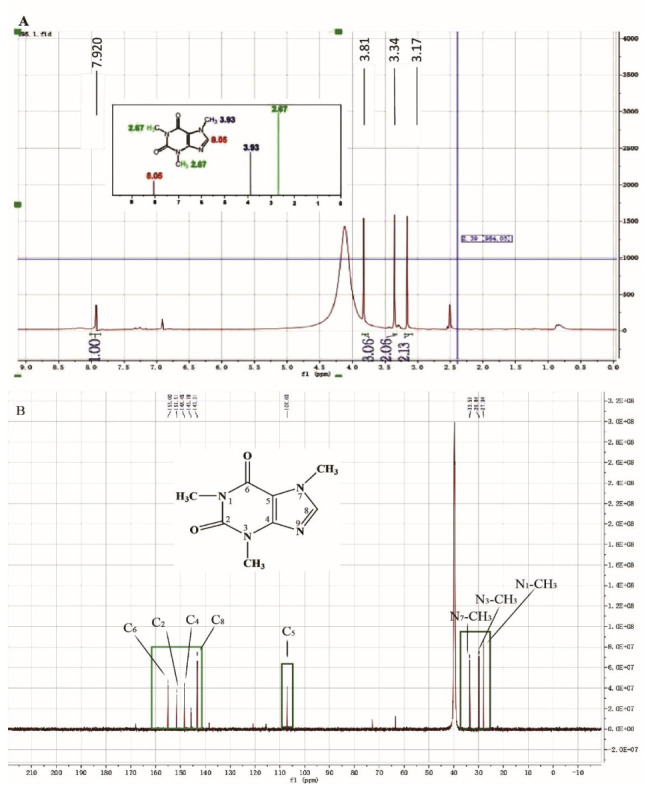
Nuclear magnetic resonance (NMR) analysis of Fu Brick tea element (**A**), carbon spectrum analysis; (**B**), hydrogen spectrum analysis.

**Figure 2 ijms-23-02525-f002:**
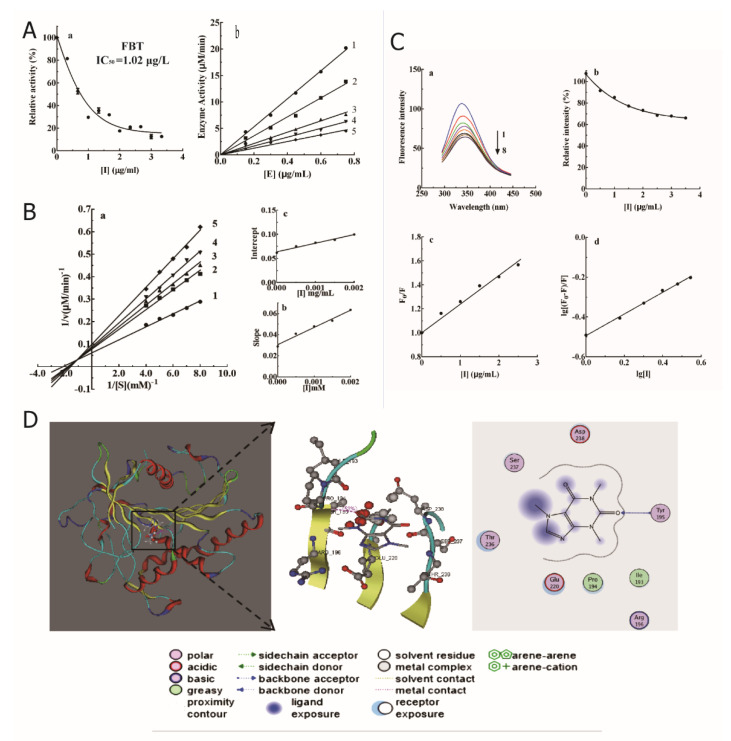
Inhibition of lipase by FBT and its mechanism. (**A**) Effect of FBT on lipase activity. (**A-a**) *IC*_50_ value of FBT; (**A-b**) Slope diagram of enzyme activity at different FBT concentrations. (**B**) Type of lipase inhibition by FBT. (**B-a**) Lineweaver–Burk double reciprocal curve; (**B-b**) double reciprocal curve slope plotted against effector concentration to determine inhibition constant *K*_I_; (**B-c**) double reciprocal curve intercept versus effector concentration plot to determine the inhibition constant *K*_IS_. (**C**) Effect of FBT on lipase emission spectrum. (**C-a**) Stratification of lipase fluorescence emission spectra; (**C-b**) Absorption peak height pair [I]; (**C-c**) Stern–Volmer curve; (**C-d**) Lg [(F_0_-F)/F] stands for LG [I]. (**D**) Molecular docking mode of FBT and lipase residues.

**Figure 3 ijms-23-02525-f003:**
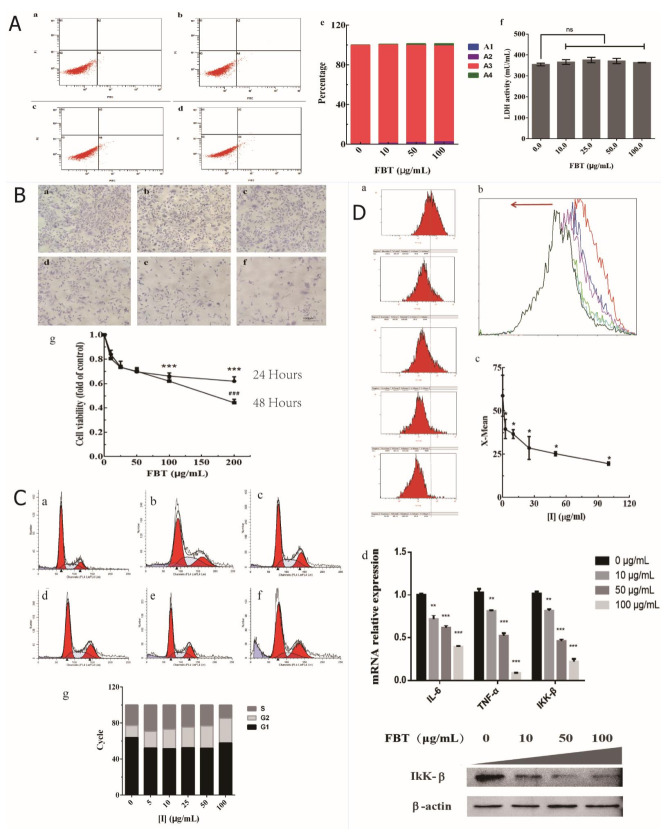
Effects of FBT on 3 T3-L1 preadipocytes. A: Effect of FBT on the apoptosis rate of 3T3-L1 preadipocytes. (**A-a**) Blank control; (**A**-(**b****–d**)) FBT dosage was 10, 50, and 100 μg/mL, respectively; (**A-e**) histogram quantization; (**A-f**) Effects of FBT on the proliferation and activity of LDH in the supernatant of 3T3-L1 cells. (**B**) Cell 3T3-L1 morphological variation after FBT treatment for 24 h. (**B-**(**a****–f**)) FBT dosage was 0, 10, 25, 50, 100, and 200 μg/mL, respectively; (**B-g**) MTT assay was used to detect the effect of FBT on the proliferation of 3T3-L1 cells. *** *p* < 0.0001 versus the control group; and ^###^ *p* < 0.0001 versus different times (24 h and 48 h) at the same concentration. (**C**) Cell cycle results. (**C**-(**a–f**)) Cells were treated with FBT at 0, 5, 10, 25, 50, and 100 μg/mL FBT for 48 h. Treated and control cells were stained with PI, and changes in the cell cycle were examined by flow cytometry. (**C-g**) Histogram showing the percentage of cell cycle distribution in each phase of the cell cycle (G0/G1, S, and G2/M). (**D**) FBT decreases intracellular ROS levels in 3T3-L1 cells; (**D-a**) flow peak diagram; (**D-b**) Cumulative displacement of ROS peaks in flow cytometry under different concentrations of FBT, where the FBT concentration gradient is 0, 5, 10, 25, 50, and 100 μg/mL; (**D-c**) X-mean quantization diagram. (**D-d**) Histogram of the influences of FBT on ROS-related genes in 3T3-L1 cells, *p* < 0.01, and *** *p* < 0.0001 versus the control group.

**Figure 4 ijms-23-02525-f004:**
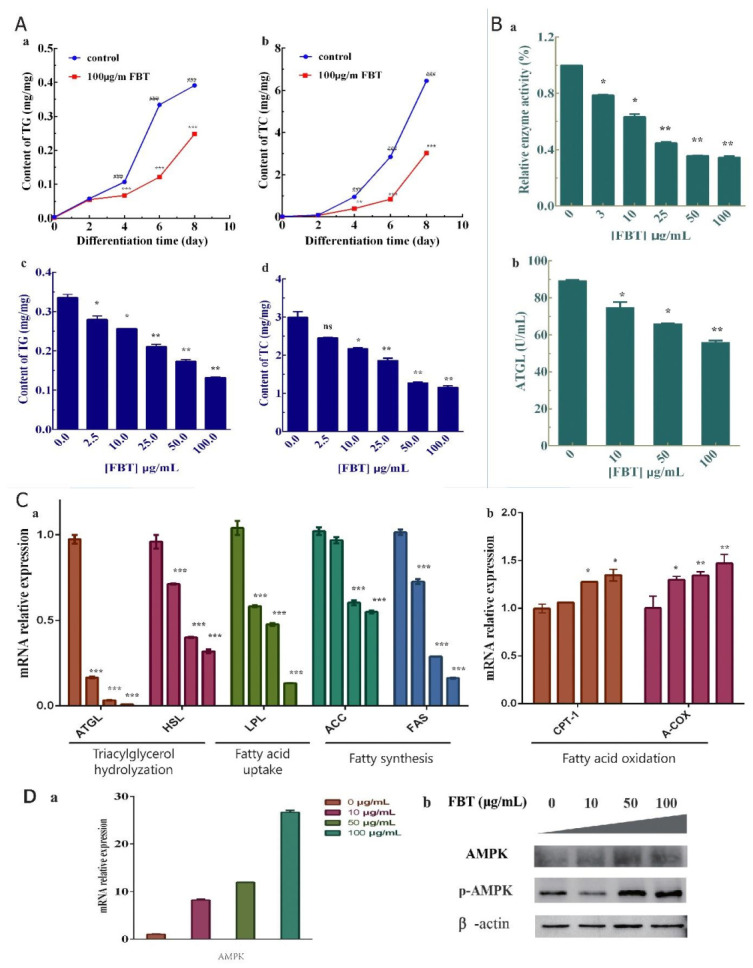
(**A**) The effect of FBT on the changes in TG and TC content during the differentiation of 3T3-L1 preadipocytes. (**A**-(**a,b**)) Changes in TG and TC contents within eight days of the cell differentiation process under the 100 μg/mL FBT treatment; (**A**-(**c,d**)) The contents of TG and TC under different concentrations of FBT on the sixth day of differentiation. * *p* < 0.05, ** *p* < 0.01, and *** *p* < 0.0001 versus the control group; ^###^ *p* < 0.0001 versus different times at the same concentration. (**B**) Effects of FBT on the related metabolism of 3T3-L1 cells. (**B-a**) 4-NPP was used as a substrate to detect lipase activity by UV spectroscopy. (**B-b**) ATGL activity detection results. (**C**) Lipid metabolism of 3T3-L1-related gene expression with FBT; (**C-a**) Gene expression related to triacylglycerol hydrolyzation, fatty acid uptake and fatty synthesis under FBT (0, 10, 50 and 100 μg/mL) treatment; (**C-b**) Effects of FBT on the mRNA levels of genes related to fatty acid oxidation. (**D-a**), AMPK gene expression with RNA; (**D-b**), AMPK and phosphorylated AMPK protein expression with Western Blot.

**Figure 5 ijms-23-02525-f005:**
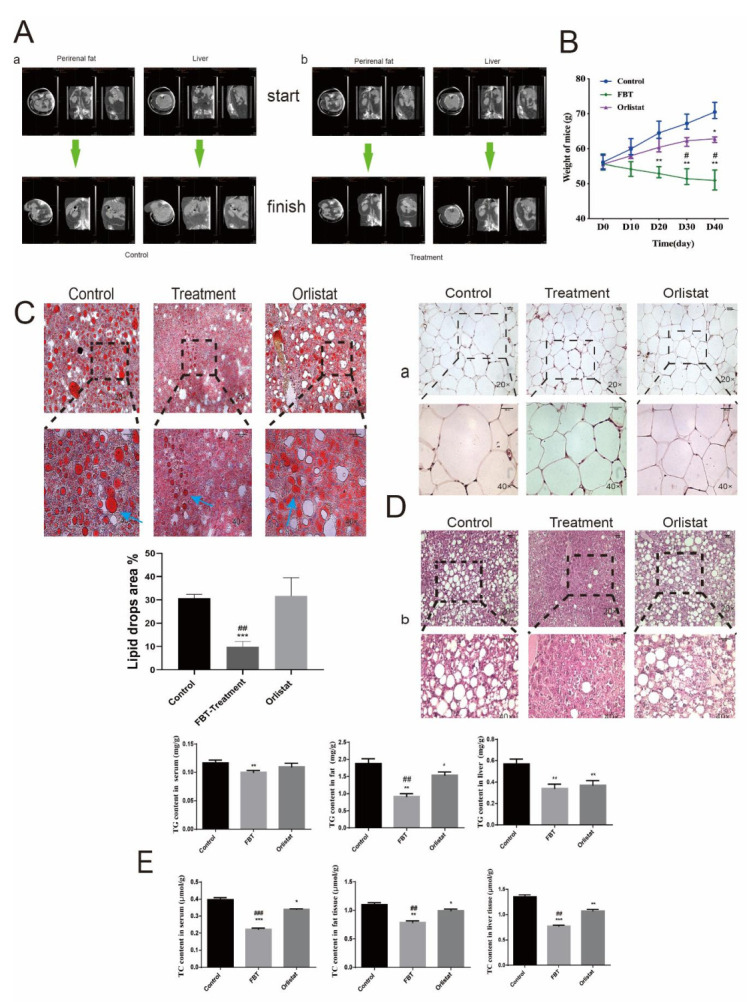
(**A**) Comparison of PET-CT scan results of mice before and after gavage. (**A-a**) PET-CT scan of perirenal fat and liver in the control group; (**A-b**) PET-CT scan of perirenal fat and liver in the FBT treatment group. (**B**) Weight changes in different groups over 40 days. (**C**) Frozen sections of liver were stained with Oil Red O, and the lipid droplet area was quantified using ImageJ. (**D**) HE staining of paraffin sections of mouse white adipose tissue and liver tissue (*n* = 6). (**E**) The contents of TG and TC in mouse liver, white fat and serum. * *p* < 0.05, ** *p* < 0.01, and *** *p* < 0.0001 versus the control group; ^#^ *p* < 0.05, ^##^ *p* < 0.01, and ^###^ *p* < 0.0001 versus the orlistat group.

**Figure 6 ijms-23-02525-f006:**
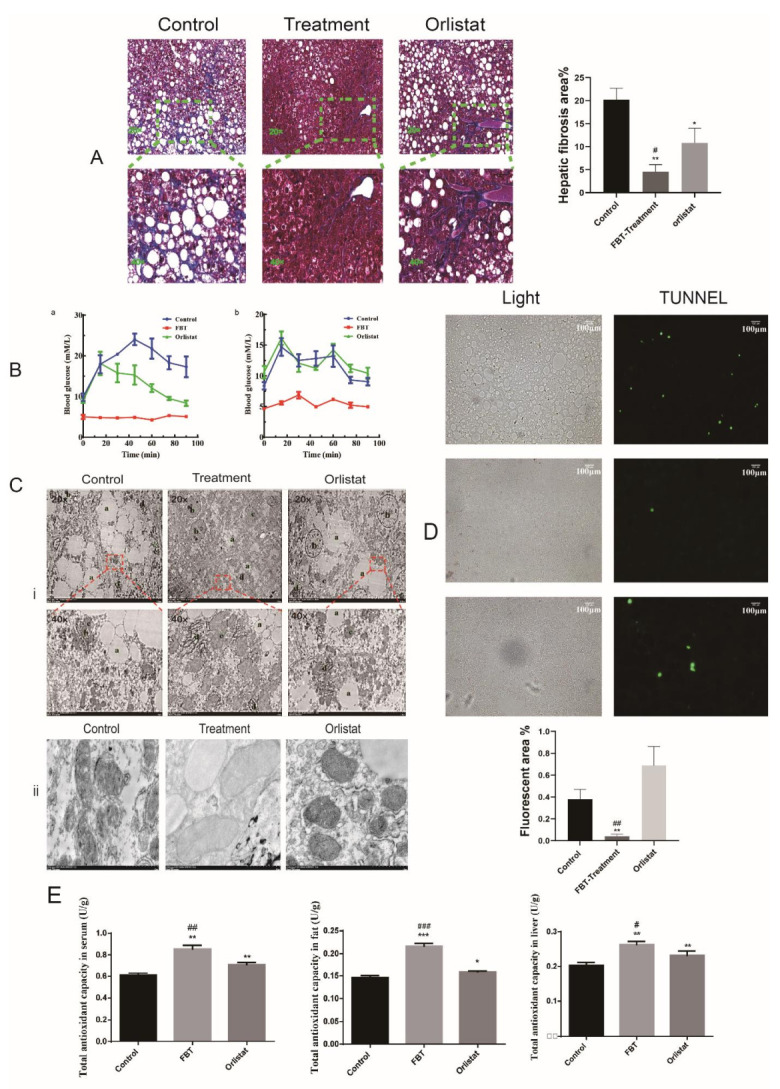
(**A**) Stained by Masson’s trichrome represents the fibrotic area, and the fibrotic area was quantified using ImageJ. The blue area is the liver tissue; The red area is the cytoplasm; The purple area is the nucleus. (**B**) Changes in blood glucose concentration within 90 min in insulin resistance and glucose tolerance tests in different groups of mice. (**C**) i; Effects of FBT on the ultrastructure of liver tissue in obese mice under a transmission electron microscope. a; lipid particles, b; nucleus, c; mitochondria. d; endoplasmic reticulum. ii; Ultrastructure of mitochondria in different treatment groups. (**D**) Mouse liver apoptosis and necrotic cells were labeled by TUNEL fluorescence staining, and the fluorescence area was quantified by ImageJ. (**E**) Detection of total antioxidant activity in the liver, fat and serum of mice. * *p* < 0.05, ** *p* < 0.01, and *** *p* < 0.0001 versus the control group; ^#^ *p* < 0.05, ^##^ *p* < 0.01, and ^###^ *p* < 0.0001 versus the orlistat group.

**Figure 7 ijms-23-02525-f007:**
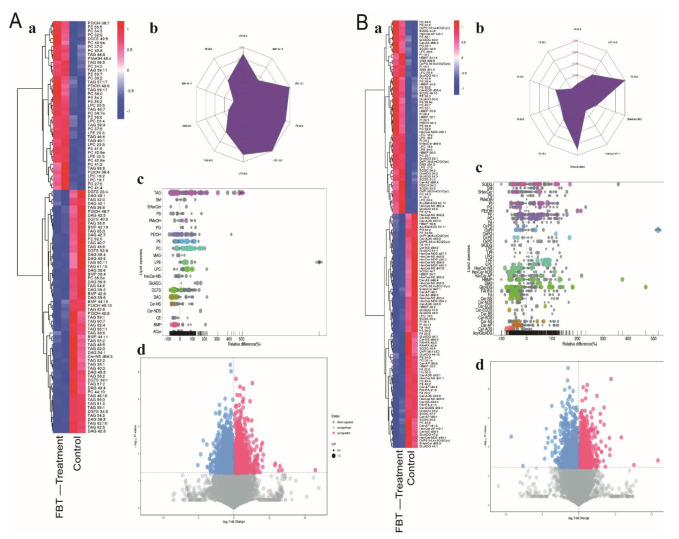
The ionization source of the LC-QTOFMS platform is electrospray ionization, and there are two ionization modes: (**A**) positive ion mode (POS), (**B**) negative ion mode (NEG). The combination of the two modes in the detection of the metabolome can increase the metabolite coverage rate. a: Heatmap of hierarchical clustering analysis for the group (FBT treatment vs. the control). b: Radar chart analysis for group. (FBT treatment vs. te control). c: Lipid species of bubble plot for the group (FBT treatment vs. the control). Each point in the lipid bubble diagram represents a metabolite. The size of the point represents the *p*-value of the Student’s *t*-test (taking the negative number of the logarithm base 10), and a larger point represents a smaller *p*-value. Grey dots represent nonsignificant differences with a *p*-value not less than 0.05, while colored dots represent significant differences with a *p*-value less than 0.05 (different colors are marked according to lipid classification). d: Volcano plot for the group (FBT treatment vs. the control).

**Figure 8 ijms-23-02525-f008:**
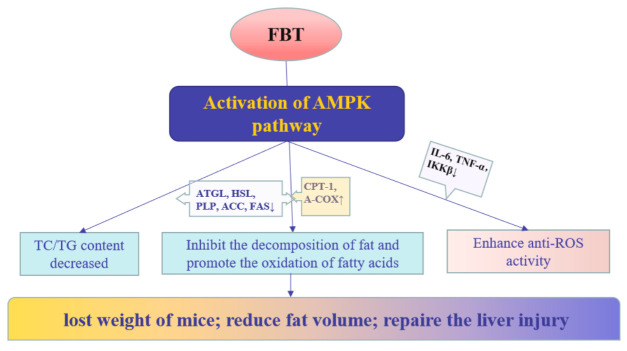
The lipid-lowering mechanism of FBT on fat metabolism in mice.

**Table 1 ijms-23-02525-t001:** Inhibition constants of FBT on lipase.

Sample	*IC*_50_ (mg/mL)		Inhibition Effect		Inhibition Constant (mM)	
			Mechanism	Types	*K* _I_	*K* _IS_
FBT	1.02	0.98	Reversible	mixed	0.93	1.78

**Table 2 ijms-23-02525-t002:** The fluorescence parameters between FBT and lipase.

Types	Quenching Type	*K*_SV_ (M^−1^)	*K*_q_ (M^−1^s)	*K*_A_ (M^−1^)	n
FBT	Static	2.39 × 10^2^	2.39 × 10^10^	3.12 × 10^3^	1.745

## Data Availability

The data presented in this study are available on request from the authors.

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
