# Peer review of "Theophylline Extracted from Fu Brick Tea Affects the Metabolism of Preadipocytes and Body Fat in Mice as a Pancreatic Lipase Inhibitor"

_ijms, 2022, doi:10.3390/ijms23052525_

Round 1

Reviewer 1 Report

The study describes the beneficial effects of the Fu Brick Tea (FBT) extract as anti-obesity drug. In preadipocyte cell line FBT acts as lipase inhibitory, reduces proliferation and differentiation by eliminating inflammation and oxidative stress.  In an animal model of  obesity, FBT reduces fatty liver and lipid metabolism due to pancreatic lipase inhibition and by reducing lipid intake. Moreover FBT prevents liver fibrosis, reduces liver cell apoptosis,  by improving mitochondrial damage and ensuring normal β-oxidation.

The experimental hypothesis seems interesting but the manuscript presents some criticisms that need to be eliminated for publication in IJMS . 

Some of the major problems of the manuscript are listed below:

  • Introduction, lines 78-80: by what results the authors conclude that “FBT promote cell glucose consumption and the production of adiponectin”. None of the showed results, according to this reviewer, underlies to these conclusions.

The results are too long because the authors also report in them explanations that instead are to be added in discussion.

  • Results, line 163: error in the measure unit .
  • Result, line 169: please correct the number of the figure
  • the Figure 3D-b is not clear, please add the legend to clarify the different groups
  • The graphs in the Figure 3A are too small and not in focus
  • Results, line 206: in the cell line the inflammation was induced?
  • Results line 220: error in the measure unit
  • The legend of the figure 4 is lacking of the description related to the expression of P-AMPK.
  • The figure 5A showing perirenal fat and hepatomegaly detected by PET-CT in mice could be enlarged. It is not clear what is described.; in the legend of 5D please add the numer of animals.
  • Results; Lines 323-325: this sentence should be described previously.
  • Figure 6Ba-b: according to this revisor the GTT and ITT show an abnormal response of the treated obese animals. In fact, after ip injection of glucose a rapid and temporary increase of glucose is usually expected also in healthy subject.
  • Results, lines 325-326: How was the antioxidant capacity detected?
  • In the Figure 6C please change the color of the letters, from green to black.
  • Discussion, lines 383-385: the sentence is repeated later. Please delete one of the two
  • Discussion, lines 397-399. According to the results the analysis of the antiinflammatory effects of the FBT was only detected in the cell line, not in animals. Please clarify this point.
  • Discussion: line 448: “…We found that FBT can effectively reduce insulin resistance and glucose tolerance..” This sentence in not supported by the results. The GTT is not enought to show insuline resistance.
  • In The Figure 8 the conclusion regarding the beneficial effects of FBT on the antiinflammation, insuline resistance antioxidant effects in the obese mice are not supported by the showed results.

Author Response

Dear Reviewer:
Thank you for your letter and for the reviewers’ comments concerning our manuscript entitled “Theophylline extracted from Fu brick tea affects the metabolism of preadipocytes and body fat in mice as a pancreatic lipase inhibitor” (ID: ijms-1552782). Those comments are all valuable and very helpful for revising and improving our paper, as well as the important guiding significance to our researches. We have studied comments carefully and have made correction, it is so appreciate for Editors/Reviewers’ warm work earnestly, and hope that the correction will meet with approval. The main corrections in the paper and the responds to the reviewer’s comments are as following:
Responds to the reviewer’s comments:

Response to Reviewer 1 Comments:

Point 1Introduction, lines 78-80: by what results the authors conclude that “FBT promote cell glucose consumption and the production of adiponectin”. None of the showed results, according to this reviewer, underlies to these conclusions.

Response 1First of all, thank you very much for the reviewer put forward this problem, it is our negligence caused the doubt to readers. This paragraph “and promote cell glucose consumption and the production of adiponectin” was related to adiponectin experiment in the original manuscript, but it was deleted at last because the data was not sufficient. Therefore, this sentence need and has been deleted in the original manuscript.

Point 2: The results are too long because the authors also report in them explanations that instead are to be added in discussion.

Response 2: The conclusion part has been appropriately deleted. Please check the revised manuscript for details.

Point 3: Results, line 163: error in the measure unit. Results line 220: error in the measure unit.

Response 3: Thank you very much for your careful correction. The error of the measure unit has been concerned corrected. Please check the revised manuscript.

Point 4: 1) Result, line 169: please correct the number of the figure. 2) the Figure 3D-b is not clear, please add the legend to clarify the different groups. 3) The graphs in the Figure 3A are too small and not in focus.

Response 4: 1) The number of the figure has corrected, now in line 209. 2) The legend of Figure 3D-b has been reedited. 3) Figure 3A has been enlarged accordingly, which mainly observes the quadrant where cells exist. It can be clearly seen that there is no significant migration of cells to quadrant 1 or 4 in the image, indicating that FBT will not lead to adipocyte apoptosis.

Point 5: Results, line 206: in the cell line the inflammation was induced?

Response 5: I'm sorry our description may not be appropriate. What we want to describe in the manuscript is that FBT can reduce the high ROS level caused by chronic inflammation due to the obesity, and has the effect of weakening inflammation rather than causing inflammation. The manuscript has been revised to more accurately describe the manuscript, see red font for details.

Point 6: The legend of the figure 4 is lacking of the description related to the expression of P-AMPK.

Response 6: Thank you very much for finding this problem. We have added the description of Figure 4D, please refer to the revised manuscript.

Point 7: The figure 5A showing perirenal fat and hepatomegaly detected by PET-CT in mice could be enlarged. It is not clear what is described.; in the legend of 5D please add the numer of animals.

Response 7: Thank you very much for the questions raised by the reviewers. We have enlarged Figure 5A, and we have added more detailed explanations about the explanation of this figure in 2.2.1, and the experimental results have been confirmed in other parts. In addition, we have added the number of mice in the legend of Figure 5D.

Point 8: Results; Lines 323-325: this sentence should be described previously.

Response 8: Thank you very much for your suggestion, we have revised this  sentence, please see the details in revised manuscript.

Point 9: Figure 6Ba-b: according to this revisor the GTT and ITT show an abnormal response of the treated obese animals. In fact, after ip injection of glucose a rapid and temporary increase of glucose is usually expected also in healthy subject.

Response 9: After insulin injection, it was found that the degree of insulin resistance in mice treated with fu Brick tea extract FBT was reduced. Although the blood glucose of the three groups did not decrease after insulin injection, they all showed strong insulin resistance. However, FBT-treated mice had significantly increased insulin sensitivity and maintained blood sugar levels. Blood glucose levels increased significantly within 15 minutes in the orlistat treatment group and the blank control group. In particular, the blood glucose in the blank control group showed a downward trend after rising to 45 minutes. Glucose tolerance decreased in the control and orlistat groups. 15min after glucose injection, blood glucose significantly increased (which should include healthy subjects), 60min to slightly decreased blood glucose, 90min back to the pre-injection level. Figure 6Ba-b shows that under the effect of FBT, the fluctuation of blood glucose level was smaller than that of the control group, and the mice in Fu Brick Tea extract group have a higher tolerance to glucose. Supplementary remarks were made in the revised manuscript.

Point 10: Results, lines 325-326: How was the antioxidant capacity detected?

Response 10: Solarbio kit was used to measure the changes of total antioxidant capacity (T-AOC) in serum, adipose tissue and liver tissue of mice. Fig.6E showed that the total antioxidant capacity of the three samples treated with Fu Brick tea extract FBT and orlistat was significantly increased. The antioxidant capacity of fatty acids in mice was increased. Thank you for your question. The relevant supplementary content has been added to the revised manuscript.

Point 11: In the Figure 6C please change the color of the letters, from green to black.

Response 11: I'm sorry that the font of the mark in the picture is too small and the color is too light make readers can't see it clearly. Now the green words have been changed to black according to the requirements of the reviewer. Thank you very much for the comments of the reviewer, which makes the picture more clear.

Point 12: Discussion, lines 383-385: the sentence is repeated later. Please delete one of the two.

Response 12: Thank you very much for the error found by the reviewer. We have removed the repeated sentence and revised the ranking of the references accordingly.

Point 13: Discussion, lines 397-399. According to the results the analysis of the antiinflammatory effects of the FBT was only detected in the cell line, not in animals. Please clarify this point.

Response 13: Thank you very much for the suggestions, following the reviewer’s comments, We had added the sentence “According to our study, anti-inflammatory effects of FBT have been detected at the cellular level and have not been detected in animals” in the discussion.

Point 14: Discussion: line 448: “…We found that FBT can effectively reduce insulin resistance and glucose tolerance..” This sentence in not supported by the results. The GTT is not enought to show insuline resistance.

Response 14: Thank you very much for your advice, we also have to consider this problem. In the earliest data, we made the insulin resistance test of cells-The effect of FBT on adiponectin of fat cell. But that's still not enough to prove the insulin resistance with FBT, after comprehensive analysis, we decided to delete this part of data, so we changed this paragraph to“We found that FBT could maintain the stability of blood glucose and insulin in mice”.

Point 15: In The Figure 8 the conclusion regarding the beneficial effects of FBT on the antiinflammation, insuline resistance antioxidant effects in the obese mice are not supported by the showed results.

Response 15: Thank you very much for your suggestion, which is very pertinent. For insulin resistance, the author initially studied the effect of adiponectin content on the cellular level and the conclusion of GTT on the animal level , but it is still not enough to prove that FBT can reduce insulin resistance. The data and content have been removed, and Figure 8 has been modified accordingly. In terms of ROS, we believe that FBT has anti-ROS activity, which is supported by flow cytometry ROS and related gene RNA results of TNF-α,IL-6, IKK-β at the cell level and total antioxidant capacity of the obese mice serum, adipose tissue, and liver tissue at the animal level.

Reviewer 2 Report

The article by Liu et al. “Theophylline extracted from Fu brick tea affects the metabolism of preadipocytes and body fat in mice as a pancreatic lipase inhibitor” provides biochemical data demonstrating that Theophylline extracted from Fu brick tea (FBT) acts as a pancreatic lipase inhibitor. The authors subsequently use two obesity models, one in vitro and the other in vivo, in order to test whether FBT is able to reduce fat accumulation, fat cell number/inflammation, and liver injury, possibly through lipase inhibition.  In the in vitro experiments (3T3-L1 preadipocytes), FBT is shown to induce a number of effects, including decrease in cell proliferation by inhibiting the exit from G2 phase in the cell cycle, reduction in lipid accumulation during preadipocyte differentiation and the activity/expression of specific adipose lipases and other enzymes involved in lipid metabolism. The in vivo FBT treatment in OB/OB mice suggest an improvement in liver steatosis and fibrosis.

The scientific question behind these experiments is interesting and bearing relevant therapeutic implications. However, the Introduction and Discussion sections lack several references to the literature and the Title does not address the fact that FBT could act through several mechanisms, including but not only pancreatic lipase inhibition.  The figures are difficult to read and the language used in the article is often unclear and the terminology imprecise.

Major concerns:

  • The Abstract is incomplete, for example the experiments on extraction of theophylline from Fu Brick Tea are not reported
  • The type of lipase employed in most experiments is not specified: is it always pancreatic lipase? From what species?
  • Many graphs and figures in general are very small, with low resolution and therefore difficult to read
  • Line 155: “The lipid-lowering effects of lipase inhibitors was verified at the cellular level” is unclear and/or incorrect.  The results obtained in FBT-treated 3T3-L1 cells could be due to mechanisms other than a lipase inhibitor effect.  Actually, the effects of (pancreatic) lipase inhibitors have been mainly demonstrated in vivo, where they reduce the adsorption of lipids through the intestine. The effect on preadipocyte proliferation/cell cycle might depend on other mechanisms.
  • Figure 3Ba-f: it would be more meaningful to show the pictures of cells treated with the same concentrations used for the MTT assay (Fig 3Bg)
  • Fig 3Dd: the lower panel (expression of Ikk-beta) shows the image of a blot or a gel? mRNA or protein? There is no description in the legend
  • Figure 7 is not readable
  • Lines 79 (Introduction) and 529 (Methods) mention experiments measuring glucose consumption and adiponectin production in 3T3-L1 cells. Where are the data?

Other recommendations:

  • FBT in most literature is used for “Fu Brick Tea”, not for “Fu Brick Theophylline”. Furthermore, there is a discrepancy in the text: in line 38 FBT stands for “Fu Brick Tea”, while in line 15 and in the rest of the article stands for “Fu Brick Theophylline”. Please modify.
  • Line 29: LPL usually stands for “Lipoprotein Lipase”, not “Lipoprotein Esterase”
  • Line 65: the authors could add their own review (Liu et al. 2020, Lipase Inhibitors for Obesity: A Review) as a reference here
  • Line 157: preadipocytes, not proadipocytes
  • Line 183: D-(a-f)  should be moved to line 186
  • Line 232: lipase content or lipase activity?
  • Lines 343-345: “the accumulation of a large amount of diglyceride (DAG), fatty acid (FA) and acetylated 343 fatty acid (FAHFA) in the liver tissue of mice without FBT treatment may be due to the  inhibition of lipase in the intestine by FBT” this sentence is not clear or not correct (without or with FBT?)
  • Line 518: adherence to “medium”?
  • Line 519: “parallels” means “replicates”?

Author Response

Point 1The Abstract is incomplete, for example the experiments on extraction of theophylline from Fu Brick Tea are not reported

Response 1Thank you very much for your accurate correction. We have added this sentence " In this study, lipase inhibitor FBT was screened from natural products based on enzyme molecular dynamics, which was extracted and purified from Fu Brick Tea. " to the abstract, please check the revised manuscript for details.

Point 2: The type of lipase employed in most experiments is not specified: is it always pancreatic lipase? From what species?

Response 2: Thank you very much for the reviewer put forward this problem, it is our negligence caused the doubt to readers. Lipase from Mucor was used in our study. As detailed in the previous paper "Inhibitory Mechanism and Molecular Analysis of Furoic Acid and Oxalic Acid on Lipase", we have done the sequence alignment in the early stage for Mucor with rat/mice/human/cattle, found that their active sites were same and made by three residue Ser, His and Asp. And be a disguise by "alpha screw top", formed by the exposure of hydrophobic group surrounded, hydrophilic group of lipase electrophilic area (oxygen anion hole). Microbial had high production quantity of enzyme, our lab had done the research of putting out lipase from Mucor, Therefore, considered the economic considerations, we chose Mucor as the source of lipase.

Point 3: Many graphs and figures in general are very small, with low resolution and therefore difficult to read.

Response 3: Thank you very much for your careful correction. Figure 3 and Figure 6 have been re-uploaded after adjustment, and the fonts have been enlarged accordingly. In addition, all images are adjusted through an AI editor to keep the original high resolution and have separate uploads.

Point 4: Line 155: “The lipid-lowering effects of lipase inhibitors was verified at the cellular level” is unclear and/or incorrect. The results obtained in FBT-treated 3T3-L1 cells could be due to mechanisms other than a lipase inhibitor effect. Actually, the effects of (pancreatic) lipase inhibitors have been mainly demonstrated in vivo, where they reduce the adsorption of lipids through the intestine. The effect on preadipocyte proliferation/cell cycle might depend on other mechanisms.

Response 4: Prophase study has certain FBT on lipase activity and achieve lipid-lowering effect, this is analyzed to explore the mechanism of molecular enzymology level, then we will be extracted from cellular level validation lipase inhibitor lipid-lowering effect, and from cells in vitro experiment further discusses and analyzes the mechanism of inhibition. As we know, the most obvious feature of obesity is the increase in body size and weight, which is caused by the excessive increase in the volume and number of fat cells in the body, resulting in the excess of adipose tissue. Changes in the number and volume of fat are determined by the proliferation, differentiation and apoptosis of proadipocytes .Therefore, it is necessary to study the effects of FBT on the proliferation, differentiation and apoptosis related regulatory genes and proteins of proadipocytes.

Point 5: Figure 3Ba-f: it would be more meaningful to show the pictures of cells treated with the same concentrations used for the MTT assay (Fig 3Bg)

Response 5: I'm sorry for the incorrect description in Figure 3Bg, which has been corrected now. We use the same concentration, and the specific detection steps are as follows: The 3T3-L1 preadipocytes were cultured to logarithmic stage to detect the effect of FBT on cell proliferation. The culture medium was abandoned and washed with PBS twice. After trypsin digestion, fresh culture medium was added for appropriate dilution. After the wall was attached, the old culture medium was aspirated, and the drug was added according to the concentration of 0, 10, 25, 50, 100, 200 μg/mL. Three groups with each concentration were parallel. After 24 h and 48 h, 20 µL MTT solution was added to each well from light, and then the medium was slightly shaken and incubated for 4 h. 150 µL DMSO solution was added to each well, and the detection was carried out at 570 nm after shock from light. The inhibition rate was calculated according to Formula.

Inhibition Rate (%) = (ODC-ODT) /ODc*100

ODC is the absorption value of the control group. ODT is the absorbance value of the dosing group.

Point 6: Fig 3Dd: the lower panel (expression of Ikk-beta) shows the image of a blot or a gel? mRNA or protein? There is no description in the legend.

Response 6: Thank you very much for finding this problem. We have added the description of Figure 3D-d, please refer to the revised manuscript.

Point 7: Figure 7 is not readable

Response 7: Thank you very much for the questions raised by the reviewers. We have re-uploaded Figure 7, please check whether it is readable.

Point 8: Lines 79 (Introduction) and 529 (Methods) mention experiments measuring glucose consumption and adiponectin production in 3T3-L1 cells. Where are the data?

Response 8: First of all, thank you very much for the reviewer put forward this problem, it is our negligence caused the doubt to readers. This paragraph in Lines 79“and promote cell glucose consumption and the production of adiponectin” was related to adiponectin experiment in the original manuscript, but it was deleted at last because the data was not sufficient. Therefore, this sentence and method section related content has been deleted in the original manuscript.

Point 9: FBT in most literature is used for “Fu Brick Tea”, not for “Fu Brick Theophylline”. Furthermore, there is a discrepancy in the text: in line 38 FBT stands for “Fu Brick Tea”, while in line 15 and in the rest of the article stands for “Fu Brick Theophylline”. Please modify.

Response 9: Many thanks to reviewer for the kind suggestion, this manuscript still wishes to use FBT as an abbreviation for Fu Brick Theophylline and correct the error in line 38.

Point 10: Line 29: LPL usually stands for “Lipoprotein Lipase”, not “Lipoprotein Esterase”

Response 10: Thank you very much for your careful correction. The errors have been corrected in the revised manuscript.

Point 11: Line 65: the authors could add their own review (Liu et al. 2020, Lipase Inhibitors for Obesity: A Review) as a reference here

Response 11: Thank you very much for the suggestions of reviewers. The references have been added, Thank you again for reading the author's article carefully.

Point 12: Line 157: preadipocytes, not proadipocytes; Line 183: D-(a-f) should be moved to line 186;

Response 12: Thank you very much for the error found by the reviewer. We have corrected the error and marked it in red. Please check in the revised manuscript.

Point 13: Line 232: lipase content or lipase activity?

Response 13: Thank you very much for the question, the sentence “ATGL kit was used to detect the decreased triglyceride lipase content” should be “ATGL kit was used to detect the decreased triglyceride lipase activity”. The error had been corrected and marked it in red. Please check in the revised manuscript.

Point 14: Lines 343-345: “the accumulation of a large amount of diglyceride (DAG), fatty acid (FA) and acetylated 343 fatty acid (FAHFA) in the liver tissue of mice without FBT treatment may be due to the  inhibition of lipase in the intestine by FBT” this sentence is not clear or not correct (without or with FBT?)

Response 14: With FBT treatment, FBT can significantly reduce the types of triglyceride TG in mouse liver tissue and prevent the generation of degenerative lipids. We have adjusted the sentence of this paragraph to make it clearer to understand..

Point 15: Line 518: adherence to “medium”?

Response 15: Yes.

Point 16: Line 519:“parallels” means “replicates”?

Response 1: Yes. Thank you very much for the reviewer put forward this problem. That means we use the same concentration gradient.

Thank you and all the reviewers for the kind advice. We look forward to hearing from you regarding our submission. We would be glad to respond to any further questions and comments that you may have.

Sincerely yours

Reviewer 3 Report

The form of the abbreviations has to be revised. I suggets to move the abbreviations at the end of the manuscript.

The abstract is too generic. Authors should rewrite the abstract to make immediate the knowledge of the major findings of the study.

Introduction: please improve the description of fu brick tea.

In material and methods, please include the name of the plant of origin. Did the author commercial or harvested sample?

Please check the use of the acronyms in the text.

Author Response

Dear Reviewer:
Thank you for your letter and for the reviewers’ comments concerning our manuscript entitled “Theophylline extracted from Fu brick tea affects the metabolism of preadipocytes and body fat in mice as a pancreatic lipase inhibitor” (ID: ijms-1552782). Those comments are all valuable and very helpful for revising and improving our paper, as well as the important guiding significance to our researches. We have studied comments carefully and have made correction, it is so appreciate for Editors/Reviewers’ warm work earnestly, and hope that the correction will meet with approval. The main corrections in the paper and the responds to the reviewer’s comments are as following:
Responds to the reviewer’s comments:

Response to Reviewer 2 Comments:

Point 1: The form of the abbreviations has to be revised. I suggets to move the abbreviations at the end of the manuscript.

Response 1: Thank you very much for the reviewer put forward this problem. The abbreviations has be revised as the order of A-Z, and be moved at the end of the manuscript.

Point 2: The abstract is too generic. Authors should rewrite the abstract to make immediate the knowledge of the major findings of the study.

Response 2: According to the comments of reviewers, we have edited the abstract again, please check it.

Point 3: Introduction: please improve the description of fu brick tea.

Response 3: Thank you very much for the suggestions, the description of fu brick tea has been improved in the section of introduction.

Point 4: In material and methods, please include the name of the plant of origin. Did the author commercial or harvested sample?

Response 4: The detailed information of the material have been added. The tea material purchased from Shannxi Jingyang Yichang Mingfu Brick Tea Co., LTD.

Point 5: Please check the use of the acronyms in the text.

Response 5: Thank you very much for your advice, the acronyms throughout have been carefully examined.

Thank you and all the reviewers for the kind advice. We look forward to hearing from you regarding our submission. We would be glad to respond to any further questions and comments that you may have.

Sincerely yours

Round 2

Reviewer 2 Report

I thank the authors for the quick and careful response.  The manuscript has been improved in several aspects, however I still have two requirements:

  • The abstract is now longer and more detailed (as requested), but the new additions need to be English edited to make them more understandable.
  • I appreciated your clarification on what type of lipase was used in the experiments.  However, it is important to add this information in the appropriate sections of the manuscript, in particular in the Materials & Methods. Please specify the source (Mucor) and add the reference of your previous paper in which you explain the rational of using this mucor enzyme instead of a rodent/human enzyme.  Also, please add the reference mentioned in line 558 “The inhibition rate, type and mechanism were measured using the method we previously described”.

Author Response

Point 1: The abstract is now longer and more detailed (as requested), but the new additions need to be English edited to make them more understandable.

Response 1: Thank you very much for your comments. Our articles have been polished by the native language of professional companies. And some grammatical errors had been corrected in the article and reedited the abstract section.

Point 2: I appreciated your clarification on what type of lipase was used in the experiments. However, it is important to add this information in the appropriate sections of the manuscript, in particular in the Materials & Methods. Please specify the source (Mucor) and add the reference of your previous paper in which you explain the rational of using this mucor enzyme instead of a rodent/human enzyme. Also, please add the reference mentioned in line 558 “The inhibition rate, type and mechanism were measured using the method we previously described”.

Response 2:Thank you very much for your comments and suggestions. In order to enhance readers’ understanding, we have added the lipase source and the specify reference in section 4.1. In addition, references have been added to this sentence "The inhibition rate, type and mechanism were measured using the method we previously described".
